Graph-based methods for Author Name Disambiguation: a survey

De Bonis Michele michele.debonis@isti.cnr.it 1
Falchi Fabrizio 1
Manghi Paolo 1 2
1 Istituto di Scienza e Tecnologie dell’Informazione ”A. Faedo” (ISTI), Consiglio Nazionale delle Ricerche (CNR) , Pisa , Italy
2 OpenAIRE AMKE , Marousi (Athens) , Greece
Bolshoy Alexander
Electronic publication date: 2023 Sep 11
Publication date: 2023
Volume: 9
Electronic Location ID: e1536
Received 2023 Mar 22; Accepted 2023 Jul 24
Copyright: ©2023 De Bonis et al.
Copyright year: 2023
Copyright holder: De Bonis et al.
License: This is an open access article distributed under the terms of the Creative Commons Attribution License, which permits unrestricted use, distribution, reproduction and adaptation in any medium and for any purpose provided that it is properly attributed. For attribution, the original author(s), title, publication source (PeerJ Computer Science) and either DOI or URL of the article must be cited.
License URL: https://creativecommons.org/licenses/by/4.0/

Keywords: Disambiguation, Deduplication, Author name disambiguation

Funding: The EU H2020 projects OpenAIRE-Nexus 101017452 EOSC-Future 101017536 This work was co-funded by the EU H2020 projects OpenAIRE-Nexus (Grant agreement ID: 101017452) and EOSC-Future (Grant agreement ID: 101017536). There was no additional external funding received for this study. The funders had no role in study design, data collection and analysis, decision to publish, or preparation of the manuscript.

==============================
Scholarly knowledge graphs (SKG) are knowledge graphs representing research-related information, powering discovery and statistics about research impact and trends. Author name disambiguation (AND) is required to produce high-quality SKGs, as a disambiguated set of authors is fundamental to ensure a coherent view of researchers’ activity. Various issues, such as homonymy, scarcity of contextual information, and cardinality of the SKG, make simple name string matching insufficient or computationally complex. Many AND deep learning methods have been developed, and interesting surveys exist in the literature, comparing the approaches in terms of techniques, complexity, performance, etc. However, none of them specifically addresses AND methods in the context of SKGs, where the entity-relationship structure can be exploited. In this paper, we discuss recent graph-based methods for AND, define a framework through which such methods can be confronted, and catalog the most popular datasets and benchmarks used to test such methods. Finally, we outline possible directions for future work on this topic.

Introduction

Scholarly knowledge graphs (SKGs) are knowledge graphs representing research-related information, such as publications, datasets, software, authors, organizations, projects, funders, and data sources. Relevant examples are Google Scholar (http://scholar.google.com), OpenAlex (http://openalex.org) (Priem, Piwowar & Orr, 2022), Semantic Scholar (http://semanticscholar.org), OpenAIRE Graph (http://graph.openaire.eu) (Manghi et al. (2019)), OpenCitations.net (http://opencitations.net) (Peroni & Shotton, 2020), etc. Researchers and service providers use SKG data to perform investigations in bibliometrics, Science of Science, etc. or offer end-user functionalities for discovery, assessment, statistics, and trends.

SKGs are populated by aggregating metadata from distinct data sources whose content typically overlaps, giving life to redundant information spaces; e.g., multiple metadata records can describe the same scientific article or the same author. Accordingly, the challenge of entity disambiguation becomes crucial to ensure high-quality and functional SKGs.

One of the main challenges in the context of disambiguating graph data is author name disambiguation (AND). Registries for researchers, such as ORCID.org, address the issue by assigning unique and persistent identifiers to researchers and disambiguating their authorship across different publications. However, they still encounter problems concerning coverage and proper and pervasive usage in bibliographic metadata, as described in Baglioni et al. (2021). De facto, in many cases, author names come without a researcher identifier, as strings provided as properties/attributes of bibliographic metadata about scientific outputs, e.g., publications, datasets, software, etc. Understanding when two author names are equivalent, i.e., refer to the same real-world person, is a task as important as complex. Simple string matching is insufficient to establish if a pair of names refer to the same author, as homonyms may lead to incorrect conclusions. Similarly, different string names may not imply that the related authors are different, as synonyms are also common, e.g., names in different languages and different name representations (“Steven Smith”, “S. Smith”, “Smith, Steven”, “Smith, S.”). To cope with homonymy and synonymy challenges, author names are typically enriched with contextual information extracted or inferred from the bibliographic metadata records where the name originally occurred; examples are titles, abstracts, subjects, dates, or topics.

Several methods have been proposed and engineered in the literature to solve the AND task, characterized by the challenges of computational complexity and author name match.

Computational complexity is typically dealt with via a preliminary clustering phase on author names to group potentially equivalent ones. Such phase is usually performed using the LN-FI (Last Name-First Initial) method, which hashes every author’s name with a string that is a concatenation between the author’s last name and the first letter of the name (i.e., “Sandra Smith” and “S. Smith” are encoded like “smiths”). Authors sharing the same hash will end up in the same group.

Author name matching, in the vast majority of cases, is based on bibliographic metadata-related information to be extracted via heuristics or inferred via AI techniques, such as decision rules or deep learning. The idea behind AI techniques is to characterize bibliographic metadata records in so-called “embeddings” and use them to create clusters of equivalent author names. In the last decade, a vast amount of scholarly data on the Web has been shared in the form of graph data, with SKGs popularity increasing every day. In such context, solutions rely on the underlying graph structure and extract author representations that encode both semantic and relational features out of the relationships (e.g., author-publication, author-dataset, publication-dataset, author-organization) and nodes (e.g., authors, publications, organizations, venues) of the graph.

Moreover, the identification of the proper benchmark becomes crucial in this context. As a matter of fact, the features and quality of the initial SKG define the boundaries of the deep learning AND techniques. Besides, using a common benchmark makes the quality and effect of the methods comparable with other results. Accordingly, researchers tend to start from known SKGs and pick them depending on the data required to apply their disambiguation strategies. Often, the benchmark is then enriched via pre-processing to ensure that information required for the specific AND methods is materialized.

Following the rich literature on AND solutions, many surveys exist, investigating and classifying methods from different perspectives. Nevertheless, none of the resulting taxonomies faces the problem from the point of view of graph-based approaches. This work reviews the literature in this specific domain, defines a framework to describe (and compare) graph-based AND methods, and sheds some light on the research directions in this domain. More specifically, the main contributions of this survey are:

• The review of the most popular graph-based AND methods;

• The definition of a framework to describe (and compare) graph-based AND methods;

• The list of SKG benchmarks used by the surveyed methods.

This paper is organized as follows: ‘Related work’ presents a brief review of all the existing surveys on the AND topic to highlight their characteristics and differences making this survey necessary; ‘Survey Methodology’ describes the methodology used to select articles on graph-based AND methods; ‘Detailed Review of Graph-Based and Methods’ presents a description of the state-of-the-art graph-based AND methods together with a description of the graph benchmarks that are commonly adopted; ‘Discussion’ defines a taxonomy for graph-based AND methods reviewed by this work, presenting a general structure for a method aimed to solve such task; ‘Conclusions’ concludes the paper proposing future works and directions for this field of study.

Related Work

This section discusses the surveys on AND methods available in the literature for motivating the need for a specific survey on graph-based AND methods.

Elliott (2010) presents a list of methods for the AND created between 2004 and the beginning of 2010. The authors highlight a list of challenges that an AND approach has to face (i.e., homonymy, name changes for marriage, spelling variations, incomplete metadata, and the impossibility of manually tagging all the authors) and classify methods in two subcategories: manual and automatic disambiguation. The two classes engage in a trade-off between precision and scalability: manual approaches are more precise but not scalable, while automatic ones show a higher error rate but can be applied to very large data collections. The reported automatic disambiguation methods are based on clustering and supervised learning. Clustering methods use publication attributes to create embeddings and cluster them to identify the work of a specific author; supervised learning methods can be categorized as naive Bayes models (Vikramkumar & Trilochan, 2014) to calculate the probability of a pair of authors to be the same person, and SVM models (Evgeniou & Pontil, 2001) trained to discriminate authors. Since the survey is one of the first on this topic, graph-based methods have not been deeply studied, and no specific method has been developed in that direction.

Ferreira, Gonçalves & Laender (2012) propose a taxonomy for characterizing the current AND methods described in the literature. The survey categorizes automatic methods based on two features: the evidence explored in the disambiguation task (web information, citation relationships, etc.) and the main type of exploited approach (author grouping methods, and author assignment methods). The author grouping methods aim at identifying groups of author names in a set of references based on properties of the publication nodes and potential relations with other nodes. They are based on similarity functions that compare two author names by using predefined techniques (Levenshtein, Jaccard, etc.), learned from ground truth data (providing pairs of equivalent/different authors), or graph-based techniques (similarity degree of two authors based on co-authors or properties of the related publications). From the resulting equivalent name pairs, such methods identify the groups of equivalent author names. Author assignment methods are instead based on the assumption that a set of references with disambiguated author names exist, i.e., classification strategies, and/or a mathematical representation of the authors exist, i.e., clustering strategies. Classification strategies, given a set of references, predict the author of the references among a set of predefined authors. Clustering strategies attempt to directly assign references to authors’ work by optimizing the fit between a set of references to an author and mathematical models used to represent that author. The survey touches on graph-based methods but only from the point of view of graph-based similarity functions.

Hussain & Asghar (2017) divide AND methods into five categories: supervised, unsupervised, semi-supervised, graph-based, and heuristic-based. Supervised techniques in this domain are based on labeled training data that associates the corresponding author record class to the author representations (e.g., embeddings, vectors). Unsupervised techniques are those adopted when labeled data is not available, but the idea is similar to the supervised methods because the method has to classify authors into a pre-defined set of classes. Heuristic-based AND techniques are used when scalability issues occur. Such methods approximate the solution giving a result that is as close as possible to reality. As for the Graph-based AND techniques, the focus is on methods relying on graphs in which author names are nodes, and edges identify the co-authors relations (when two author names occur in the same publication). Similarity measures or deep learning is then applied to such graph to identify groups of equivalent author names.

Shoaib, Daud & Amjad (2020) provide a generic five steps framework to handle AND issues. Such steps are (i) dataset preparation, (ii) publication attributes selection, (iii) similarity metrics selection, (iv) model selection, and (v) clustering performance evaluation. An important contribution of this survey is the definition of a set of common challenges in the AND to be faced when developing a framework. The methods in this survey are categorized into supervised, unsupervised, semi-supervised, graph-based, and ontology-based. Classes defined in the taxonomy resulting from this survey have already been described by other surveys reviewed in this paragraph. As for ontology-based classification, the survey defines ontology as the knowledge of concepts and their relationships within a domain, i.e., the knowledge representation of a domain. Methods included in this category take advantage of such representation to identify groups of equivalent author names.

Sanyal, Bhowmick & Das (2021) focus on AND challenges in PubMed (https://pubmed.ncbi.nlm.nih.gov), the publication repository of the Life Science community. This work surveys a set of solutions considering as input data the citation graph formed by PubMed where the author set is not known a priori. The outcome is a general framework composed of four stages: (i) citation extraction, (ii) LN-FI blocks creation, (iii) similarity profile creation (creation of similarity for each pair of citations), and (iv) author-individual clusters creation based on similarity profiles. The survey proposes a taxonomy that classifies methods based on evidence explored (only co-authorship information or multiple metadata), and techniques used to generate similarity profiles (supervised, graph-based, and heuristic-based). Only one graph-based method is described in this review, named GHOST (GrapHical framewOrk for name diSambiguaTion), also included in Hussain & Asghar (2017) above. GHOST, presented in Fan et al. (2011), implements a similarity measure based on the graph composed of co-authors relations and applies a clustering method to create groups of equivalent author names.

A recap of the characteristics of each survey described in this section is depicted in Table 1. Clearly, in recent years, graph-based approaches are regarded as relevant but only as one of the potential classes of AND solutions; no specific investigation digs into the features of this class of problems and methods.

Table 1 Summary of existing surveys on AND methods.

Survey	Taxonomy	
Elliott (2010)	manual and automatic disambiguation	
Ferreira, Gonçalves & Laender (2012)	evidence explored (web information, citation information, and implicit evidence) or exploited approach (author grouping methods, and author assignment methods).	
Hussain & Asghar (2017)	supervised, unsupervised, semi-supervised, graph-based, and heuristic-based	
Shoaib, Daud & Amjad (2020)	supervised, unsupervised, semi-supervised, graph-based, and ontology-based	
Sanyal, Bhowmick & Das (2021)	based on evidence explored or techniques used to generate similarity profiles (graph-based, heuristic-based, supervised)	

Survey methodology

The articles included in this survey have been identified by searching Google Scholar for the keywords “graph based author name disambiguation”. To prioritize the latest research trends in the domain, we limited our candidates to work published after 1/1/2021. Since we observed that after 100 search results, topics tended to diverge from our focus, we interrupted the investigation there.

To ease our investigation process (and for the reproducibility of this survey), the search was performed using a Python script (https://github.com/WittmannF/sort-google-scholar) from Wittmann (2017) with the following command:

$  python  sortgs . py -- kw  ”graph  based  author  name  disambiguation”  --startyear  2021

The command returns a CSV file that contains the first 94 publications matching the query (articles with corrupted metadata have been excluded), each with metadata about Title, Number of Citations, and Rank; by default, the script sorts the list by number of citations. The CSV has been explored to identify work relevant to our survey: papers were downloaded, first selected based on the relevance of the abstract, and then studied; the number of citations and the query rank were used as an indicator of quality but not as selection criteria; the reference list of selected articles was also analyzed to identify further relevant titles published before 2021. The full list of articles returned by the query on the 1st of July 2023 is available as a CSV in: Michele De Bonis. (2023). List of articles resulting from the Google Scholar search “graph based author name disambiguation” published after 1/1/2021 [Data set]. Zenodo. 10.5281/zenodo.8117573. The CSV includes a column to show the ones selected for this survey (including 1 article published in 2019 identified exploring the bibliographies); note that three articles are marked as “N.A.”, as they matched our relevance criteria but are written in Chinese, and no English version could be found.

Detailed Review of Graph-Based AND Methods

In this section, we present a review of methods proposed in the literature to solve the AND task when disambiguating authors in SKGs. As previously stated, we focused our analysis on graph-based methods because such methods have been proven to be very effective and are becoming the most popular in this topic. Therefore, a deep study of the methods to highlight differences and similarities could interest the research community. As an extension of this analysis, we have described and analyzed the most popular datasets and benchmarks used to measure the performance of an AND method.

Methods

LAND. The work in Santini et al. (2022) proposes a framework called Literally Author Name Disambiguation (LAND), a representation learning method without training data. Such framework utilizes multimodal literal information generated from the SKG to create node embeddings for the graph (so-called Knowledge Graph Embeddings, KGEs). The authors conducted experiments on a graph containing information from Scientometrics Journal from 1978 onwards (OC-782K), and a graph extracted from a well-known benchmark for AND provided by AMiner (AMiner-534K). LAND is based on three components:

• Multimodal embeddings: learn representative features of entities and relations in the graph by using a multimodal extension of a semantic matching model called DistMult. The extension is based on literals of publication attributes (e.g., publication title encoded with SPECTER from Cohan et al. (2020)—a pre-trained BERT model, and date encoded as described in LiteralE) which are used to modify the scoring function in order to maximize the score for existing triples and minimizing the scores for non-existing triples;

• A blocking procedure: divides authors in groups with LN-FI blocking to reduce the number of pairwise comparisons required by the AND task;

• Clustering: Hierarchical Agglomerative Clustering (HAC) presented in Müllner (2011), used to split embeddings in the same block into k-clusters that identify each unique author.

HGCN. Qiao et al. (2019) propose a novel, efficient, re-trainable and incremental AND framework based on unsupervised learning since it does not need labeled data. The idea is to construct a publication heterogeneous network for each ambiguous name using the meta-path approach. Such publication network is consequently processed by a custom heterogeneous graph convolutional network (so-called HGCN) that calculates the embeddings for each node encoding both graph structure and node attribute information. Once all publication embeddings have been computed, authors use a graph-enhanced clustering method for name disambiguation that can significantly accelerate the clustering process without the specification of the number of distinct authors. Experiments have been performed over two datasets provided by AMiner and CiteSeeX, two widely used benchmarks in author name disambiguation. The proposed method is based on two components:

• Publication heterogeneous network (PHNet) embedding: each publication is vectorized using Doc2Vec, subsequently the HGCN aggregates the publication vectors to create the publication final embedding. The research tests different GCNs to perform the aggregation (i.e., DeepWalk by Perozzi, Al-Rfou & Skiena (2014), LINE by Bandyopadhyay, Das & Murty (2020), meta-path2Vec by Dong, Chawla & Swami (2017), Hin2Vec by Fu, Lee & Lei (2017), and GraphSAGE by Hamilton, Ying & Leskovec (2017));

• Clustering: uses graph-enhanced HAC (GHAC) over the publication graph for the ambiguous name.

AE. Xiong, Bao & Wu (2021) propose an unsupervised representation learning framework based on four modules to bridge the gap between semantic and relational embeddings. The goal of the research is to jointly encode both semantic and relations information into a common low-dimensional space for AND task. Experiments have been performed over datasets taken from AMiner, DBLP, and CiteCeerX. The four modules of the framework are:

• Semantic embedding module: publications are processed with Word2Vec by Mikolov et al. (2013) using TF-IDF weighting to represent content;

• Relationship embedding module: the framework constructs the homogeneous network applying three meta-paths (the result is a unique homogeneous network where the weight of each relationship between two publications is given by the number of meta-paths connecting those publications). Each node is embedded with a MultiLayer Perceptron (MLP) which uses triplet loss to train;

• Semantic and relationship joint embedding module: a variational autoencoder is used to learn the joint embedding by minimizing reconstruction loss;

• Clustering: HAC to create publications clusters inside a group of ambiguous authors.

jGAT. Zhang et al. (2021) present a solution that considers both content and relational information to disambiguate. In the research, authors construct a heterogeneous graph based on meta-information of publications (e.g., collaborators, institutions, and venues). The heterogeneous graph is subsequently transformed into three homogeneous graphs using three meta-paths (co-author, co-organization, and co-venue meta-path). Graph Attention Networks (GAT) presented in Veličković et al. (2017) is used to jointly learn content (abstract information and title) and relational information by optimizing an embedding vector: each node (publication) of the graph is vectorized (using Word2Vec), subsequently, an embedding is computed for each meta-path, and a concatenation of the three embeddings is inputted to a fully connected network to create the k-dimensional vector representing the final embedding of the publication. Finally, a clustering algorithm is presented to gather author names most likely representing the same person (spectral clustering algorithm to learn the embedding vectors which have been learned by GAT). Experiments for this framework are performed over AMiner.

RF-LRC. Rehs (2021) develop a robust supervised machine learning approach in combination with graph community detection methods to disambiguate author names in the Web of Science publication database. The framework uses publication pairs to train a Random Forest and a Logistic Regression Classifier. The labeled data is given by the ResearcherID, through which a pair can be identified as equal or not, and the features are properties of the pair (i.e., the result of the comparisons of publication fields). The classifier is used to create a graph which is consequently inputted to the infomap graph community detection algorithm presented in Zeng & Yu (2018) to identify all publications belonging to the same author. The distinction is always performed in a subset of publications with ambiguous authors, obtained using the LN-FI blocking strategy. Experiments and training have been performed over a set extracted from WOS.

sGCN. Chen et al. (2021) provide a disambiguation model based on GCN that combines both attribute features and linkage information. The first step consists in computing the embeddings of the publications using Word2Vec. Then three different graphs are built:

• a paper-to-paper graph: nodes of the graph are publications and an edge is drawn whenever the similarity of the attributes exceeds a threshold;

• a co-author graph: all authors are represented as nodes and an edge indicates that there is a cooperative relationship between the authors;

• a paper-to-author graph: publications and authors represented as nodes and edges representing relations between publication and author.

Each graph is fed to a specialized GCN and the final output is a hybrid feature computed following the following steps: (i) embeddings of publications and authors are obtained respectively from AuthorGCN and PaperGCN, (ii) triples samples from the paper-to-paper graph to minimize the error, (iii) triples sample from the co-author graph to minimize the error, and (iv) triple samples from paper-to-author graph to minimize the error and update network weights at the same time. The PaperGCN output is the final embedding of the publication. Finally, the HAC algorithm is applied to divide publications into disjoint clusters of authors. Experiments have been performed over datasets taken from Aminer: the AMiner-18 dataset, and the AMiner-12 dataset. In addition, authors constructed a bilingual dataset from the dataset provided by China Association for Science and Technology.

LP. Mihaljević & Santamaría (2021) present a semi-supervised algorithm to disambiguate authorship pairs: the method consists of various nonlinear tree-based classifiers trained to classify pairs of authors in order to construct a graph, which is subsequently processed with label propagation to cluster group of authors. The LN-FI blocking strategy is applied to create groups of publications with ambiguous authors, subsequently, a probabilistic classification model is trained to decide whether two publications within a given block belong to the same author. The classifier resulting from the training is used to create authorship graphs as follows: publications are represented as nodes, while an edge is drawn between two nodes if the classifier predicts that both are authored by the same person. The classifier’s class probabilities are used as edge weights to obtain a labeled graph. Finally, a clustering algorithm based on the label propagation algorithm is applied to the constructed graph. The label propagation algorithm works as follows: each node of the graph is initialized with a random unique label, then the process starts and each node is labeled iteratively with the label shared by the majority of its neighbors until an equilibrium is reached. Experiments have been performed over a dataset for author disambiguation taken from ADS.

DND. Chen et al. (2023b) present a supervised Distributed Framework for Name Disambiguation (so-called DND), developed as a linkage prediction task to overcome the limitations of knowing the number of clusters a priori. Authors of the framework train a robust function to measure similarities between publications to determine whether they belong to the same author. Publications features are transformed into vectors using Word2Vec, such publications are subsequently used as nodes in a fully connected publication network where dashed lines denote ambiguity relationships between two authors in a publication pair. Each pair of publications that have an ambiguity relation is processed by a classification task, which returns 1 if the same author writes the pair and 0 otherwise. Finally, DND merges initial partitions by a rule-based algorithm to get the disambiguation result. Experiments have been performed over two datasets: the first is the AMiner dataset, and the second is from an author disambiguation competition held by Biendata (https://www.biendata.xyz/competition/whoiswho1).

MFAND. Zhou et al. (2021) present a framework called Multiple Features Driven Author Name Disambiguation (so-called MFAND). The authors construct six similarity graphs (using the raw document and fusion feature) for each ambiguous author name. The structural information (global and local) extracted from these graphs is inputted into a novel encoder called R3JG, which integrates and reconstructs the information for an author. An author is therefore associated with four types of information: the raw document feature, the publication embedding based on the raw feature, the local structural information from the neighborhood, and the global structural information of the graph. Each node is embedded by using the Random Walk on the fusion feature graph. The goal of the framework is to learn the latent information to enhance the generalization ability of the MFAND. Then, the integrated and reconstructed information is fed into a binary classification model for disambiguation. Experiments have been performed over datasets taken from AMiner.

DHGN. Zheng et al. (2021) propose a dual-channel heterogeneous graph network (so-called DHGN) to solve the name disambiguation task. In the research, authors use the heterogeneous graph network to capture various node information to ensure the learning of more accurate data structure information. FastText presented in Bojanowski et al. (2016) is used to extract the semantic information of the data through the textual information, which generates a vector representing each publication. Then the semantic similarity matrix of the publications is obtained, by computing the cosine similarity between such vectors. On the other side, the meta-path Random Walk algorithm is used to extract the features from the publications, especially from the relationships, by computing their feature vectors, and their similarity matrix. Once both the semantic and the relationship features have been exploited, the similarity matrixes are merged to compute the similarity matrix fusion. Such matrix is clustered by means of DBSCAN, an unsupervised clustering algorithm. Experiments were performed on the AMiner-WhoisWho dataset.

SA. Pooja, Mondal & Chandra (2022) propose an approach that uses attention-based graph convolution over a multi-hop neighborhood of a heterogeneous graph of the documents for learning representations of the nodes. The approach consists of an AutoEncoder-based representation learning method divided into an encoder and a decoder. The encoder performs the following operations:

• generates the initial vectors representing the nodes;

• generates node representations based on attention over neighbor types;

• fine-tunes the node representations based on attention over different relation types.

The decoder takes the output of the encoder to generate a homogeneous graph without considering relation types. Finally, vectors coming from the decoder are clustered by using HAC. Experiments have been performed over datasets taken from AMiner.

SSP. Xie et al. (2022) propose a method based on representation learning for heterogeneous networks and clustering, and exploits the self-attention technology. The method is able to capture both structural and semantic features of a publication and uses the weighted sum of those two embeddings to cluster publications written by the same author with HAC. The structural features of a publication are extracted by using meta-paths (in particular Paper-Author-Paper, Paper-Organization-Paper, Paper-Venue-Paper, Paper-Year-Paper, and Paper-Word-Paper). The representations of publications are subsequently learned by a skip-gram model. The semantic features of a publication are extracted from the title, the abstract, and the keywords by using Doc2Vec. Experiments have been performed over the AMiner-WhoIsWho dataset and the disambiguation dataset from CiteSeerX.

Datasets and benchmarks

One of the main challenges in the literature of AND is the identification of benchmarks to measure and compare performances against other state-of-the-art methods. Many organizations that work with SKGs usually produce a custom dataset of nodes and relations to be used for testing purposes. Such datasets exploit unique author identifiers to associate each author to the related list of publications independently from the actual author names specified in the publication metadata. As such, the datasets provide both the ground truth for the AND task (e.g., identifier and related names) as well as contextual information for author names (e.g., co-authors) to be used as evidence to be explored.

The methods analyzed and revised in this survey take advantage of different benchmarks available in the literature. Most of them rely on datasets derived from AMiner (https://www.aminer.org), a free online service used to index, search, and mine big scientific data designated to identify connections between researchers, conferences, and publications.

We report below the list of the AMiner datasets used for the experiments performed in the articles reviewed in this survey:

• AMiner-WhoIsWho (Chen et al. (2023a)) is the world’s largest manually-labeled name disambiguation benchmark. Authors have released 3 versions of the dataset containing more than 1,000,000 articles;

• AMiner-534K (Santini (2021a)) is a knowledge graph extracted from an AMiner benchmark. Structural triples of the knowledge graph are split into training, testing, and validation for applying representation learning methods;

• AMiner-18 (https://github.com/neozhangthe1/disambiguation) contains a total of about 40,000 raw authors and more than 200,000 publications;

• AMiner-12 (https://www.aminer.cn/disambiguation) involves more than 1,500 raw authors and about 7,500 publications;

Another popular source for datasets is CiteSeerX (https://citeseerx.ist.psu.edu), the first search engine for academic research known for its usability in the computer engineering and informatics fields. One important contribution for the AND is given by the dataset (http://clgiles.ist.psu.edu/data) providing a collection of ambiguous names and associated citations.

The Santini (2021b) OC-782K dataset is a knowledge graph extracted from a triple store covering information about the journal Scientometrics and modeled according to the OpenCitations Data Model.

A recap of the most popular datasets and benchmarks used in the reviews is depicted in Table 2.

Other popular academic search engines providing an SKG that can be used to create new benchmarks for the evaluation of an AND framework are DBLP, (https://dblp.org) WOS (https://www.webofscience.com/wos/alldb/basic-search), ADS (https://ui.adsabs.harvard.edu/) and OpenAIRE Explore (https://explore.openaire.eu/); the latter needs a preliminary processing strategy to define a ground truth for the AND as includes ORCID identifiers as well as authors without identifiers.

Discussion

The literature analysis revealed that graph-based AND methods could be represented employing the framework depicted in Fig. 1. The framework describes different methods as instances of the same workflow template, featuring some or all of the identified steps.

SKGs are characterized by semantic features, i.e. a set of publication nodes with title, abstract, author names, venue, and publishing date, and by semantic features, i.e., the set of relationships between publication and authors, citation relationships, or other contextual information, such as relationships of publications to organizations, topics, etc. As shown in Fig. 1, the SKG is the source of information necessary to produce homogeneous node representations that will be in turn input to a node clustering module that will identify the groups of equivalent author names. Node representations are generated by the node representation module, exploiting the semantic and relational features extracted from the input SKG: via the node semantic module, i.e., extracting semantic information from publication metadata, and the relation semantic module, i.e., generating homogeneous graphs to capture semantic information from the topology of several graph views in order to provide some context to the next module. Node representations can be in the form of embeddings, generated via graph-based methods, or similarity vectors, in turn forming node similarity matrices; in other words, the essence of nodes are captured via node neighborhood strategies, with the number of hops determined by the number of layers of the network for the computation, or via similarity degrees of the node with all the nodes in the same cluster. Finally, once the node representations have been computed, they are fed to a node clustering module whose purpose is to group equivalent author names using strategies that depend on the node representation nature, e.g., embeddings distance, similarity matrix, graph cliques. Typically, clustering is limited to a set of candidate author names, identified by a blocking method that groups all publications related with names of authors that are potentially equivalent; the majority of methods adopt an LN-FI strategy, e.g., “John Smith” generates a key “smithj”.

Table 2 Recap of most popular datasets and benchmarks for AND.

dataset name	source	number of entities	number of authors	
WhoIsWho	AMiner	1,102,249	72,609	
AMiner-534K	AMiner	179,377	110,837	
OC-782K	Scientometrics	293,186	188,565	
AMiner
disambiguation
dataset	AMiner	70,258	12,798	
CiteSeerX
disambiguation
dataset	CiteSeerX	8,453	468	
AMiner-18	AMiner	203,078	39,781	
AMiner-12	AMiner	7,447	1,546	

Figure 1 General framework for graph-based AND methods in this survey.

In a more detailed view, the modules can be summarized into:

• The node semantic module processes the input SKG to compute vector representations for publications in the SKG to be provided as input to the node representation module. This module produces the so-called “raw embeddings”, calculated by means of the node attributes without considering relations. Algorithms applied within the module could be used to simply clean the data in order to prepare them for the pairwise comparison or to convert string attributes to vectors easily comparable with rather simple mathematical operations;

• The relation semantic module processes the input SKG to extract homogeneous graphs capturing a specific relational interpretation of authors or publications in the graph. A known approach is the one of “meta-paths” (exploited in Santini et al. (2022), Qiao et al. (2019), Xiong, Bao & Wu (2021), Zhang et al. (2021), Chen et al. (2021), Zhou et al. (2021), Zheng et al. (2021), and Xie et al. (2022)) which creates homogeneous graphs from the input SKG from which meaningful embeddings can be subsequently computed; for example, the application of the “co-author” meta-path or the “co-venue” meta-path approach generates respectively a graph where authors are nodes linked by relationships if they co-authored one publication or a graph where publications are related if they share the same venue. Such graphs can be used to generate different author representations, capturing relational features of the graph and the authors therein. The module can be used to generate one or more graphs and potentially these can be merged to produce compound views;

• The node representation module is the core of the framework as it processes the input of the node semantic and relation semantic modules (i.e., homogeneous graphs and/or the publication vector representations) to generate node representations to support the subsequent clustering of publications written by the same authors; a node representation captures the semantic features of an individual publication and in some cases include features of a given author; node representations can be: (i) similarity vectors obtained by similarity comparisons between pairs of publications potentially written by the same author, identified via LN-FI pre-clustering strategies on author names; (ii) node embeddings obtained via networks applied to nodes in the neighborhood of the graphs resulting by merging node and relation semantic information;

• The node clustering module applies a clustering method to blocks of publications potentially written by the same author, identified via LN-FI pre-clustering strategies on author names. Clustering algorithms differ depending on the nature of node representations, i.e., clustering functions acting on graphs for similarity vectors or clustering functions on an embedding vector space.

To better highlight the differences between the methods reviewed in this survey, we provide Table 3, where for each method we indicate how each module specifically fulfills the workflow modules described above.

Table 3 Recap of AND methods modules.

method	article	semantic module	relation module	node representation module	clustering module	
LAND	Santini et al. (2022)	SPECTER, LiteralE	N.A.	DistMult multimodel extension	HAC	
HGCN	Qiao et al. (2019)	Doc2Vec	meta-path	HGCN	Graph-enhanced HAC	
AE	Xiong, Bao & Wu (2021)	Word2Vec	meta-path	Variational AutoEncoder	HAC	
jGAT	Zhang et al. (2021)	Word2Vec	meta-path	GAT	spectral clustering	
RF-LRC	Rehs (2021)	N.A.	N.A	Random Forest & Logistic Regression Classifier	infomap algorithm	
sGCN	Chen et al. (2021b)	Word2Vec	meta-path	specialized GCNs	HAC	
LP	Mihaljević & Santamaría (2021)	N.A.	N.A.	tree-based classifier	label propagation	
DND	Chen et al. (2021a)	Word2Vec	N.A.	N.A.	rule-based algorithm	
MFAND	Zhou et al. (2021)	Random Walk	N.A.	R3JG	binary classification	
DHGN	Zheng et al. (2021)	FastText	N.A.	Random Walk	DBSCAN	
SA	Pooja, Mondal & Chandra (2022)	Word2Vec	N.A.	Spectral GCN & Dense Network	HAC	
SSP	Xie et al. (2022)	Doc2Vec	meta-path	skip-gram	HAC	

Following the comparison of the different methods, Fig. 2 identifies a taxonomy that classifies AND approaches in three macro features. A method can be characterized with respect to the learning strategy, the evidence explored, i.e., the type of information used to create the author representations, and the node representation strategy. A recap of the surveyed graph-based AND methods with respect to this taxonomy is depicted in Table 4.

Figure 2 Proposed taxonomy for graph-based AND methods.

The first and most common feature in deep learning surveys is the learning strategy, which defines the approach used to train the graph-based network. Depending on the training methodology, a method may be supervised (i.e., Rehs (2021), and Chen et al. (2023b)), unsupervised (Santini et al. (2022), Qiao et al. (2019), Xiong, Bao & Wu (2021), Zhang et al. (2021), Chen et al. (2021), Zhou et al. (2021), Zheng et al. (2021), Pooja, Mondal & Chandra (2022), and Xie et al. (2022)) and semi-supervised (i.e., Mihaljević & Santamaría (2021)). Usually, unsupervised methods include a preliminary blocking stage and subsequently an approach based on Graph Convolutional Networks (GCN) and/or AutoEncoders to vectorize graph elements by creating an embedding of the publication node. In GCN methods the embedding of a node is usually created by aggregating information from the node and its related neighbors, while AutoEncoders exploit an artificial neural network to generate efficient encodings of nodes. Supervised methods sometimes include a preliminary stage of blocking, followed by pairwise comparisons exploiting either a network or a classifier trained to recognize whether a pair of author names is equivalent. Semi-supervised approaches tend to be a mixture of the above-mentioned methods. A known challenge of such non-unsupervised approaches is the cost of producing a well-defined training set of data.

Graph-based AND approaches can be classified according to the explored evidence, intended as the type of information used to generate the embeddings. In the case of a method based on publication features, the procedure relies on publication attributes such as title, date, and abstract to represent a node. Instead, a method based on relations and publication features takes advantage of the relations between publications and, in general, nodes of the SKG. As highlighted in many AND surveys, such methods, capturing both topological and semantic representations of a node, turn out to be the most promising in the literature.

The node representation strategy feature describes the strategy used to compute node representations. Learning methods compute embeddings of the nodes by aggregating information from their neighborhood. They are typically applied on homogeneous graphs as returned from the graph processing module (e.g., the meta-path approach described above). Pairwise comparison methods (i.e., Rehs (2021), Mihaljević & Santamaría (2021), and Chen et al. (2023b)) compare nodes of the graph with others to generate similarity vectors in which each element indicates the similarity degree between the target node and the others. Depending on the method, the vector may contain 0 (different author) or 1 (same author), or a similarity degree. Such representations can be used to create a similarity graph (in some cases called “ambiguity graph”) in which clustering algorithms can identify “cliques” of nodes, e.g., groups of equivalent author names.

Table 4 Recap of graph-based AND methods.

method	article	learning strategy	node representation strategy	explored evidence	dataset	
LAND	Santini et al. (2022)	unsupervised	learning	joint relation and publication feature	AMiner-534K, OC-782K	
HGCN	Qiao et al. (2019)	unsupervised	learning	joint relation and publication feature	AMiner dataset, CiteSeerX disambiguation dataset	
AE	Xiong, Bao & Wu (2021)	unsupervised	learning	joint relation and publication feature	AMiner custom dataset, CiteSeerX disambiguation dataset, DBLP custom dataset	
jGAT	Zhang et al. (2021)	unsupervised	learning	joint relation and publication feature	AMiner disambiguation dataset	
RF-LRC	Rehs (2021)	supervised	pairwise comparisons	publication features	WOS custom dataset	
sGCN	Chen et al. (2021b)	unsupervised	learning	joint relation and publication feature	AMiner-18, AMiner-12	
LP	Mihaljević & Santamaría (2021)	semi-supervised	pairwise comparisons	publication features	ADS custom dataset	
DND	Chen et al. (2021a)	supervised	pairwise comparisons	publication features	AMiner disambiguation dataset	
MFAND	Zhou et al. (2021)	unsupervised	learning	joint relation and publication feature	WhoIsWho	
DHGN	Zheng et al. (2021)	unsupervised	learning	joint relation and publication feature	WhoIsWho	
SA	Pooja, Mondal & Chandra (2022)	unsupervised	learning	joint relation and publication feature	AMiner disambiguation dataset	
SSP	Xie et al. (2022)	unsupervised	learning	joint relation and publication feature	WhoIsWho & CiteSeerX disambiguation dataset	

Conclusions

Graph-based methods proved to be more efficient than the traditional approaches in the literature and are today trending in this area of research. In this survey, we reviewed the most popular graph-based AND methods, also providing a brief recap of the datasets and benchmarks they used. The survey yields a twofold contribution: (i) an AND framework, identifying the modules and their interactions constituting a generic AND method, and (ii) a three-class taxonomy that characterizes methods based on learning strategy (common to all surveys), explored evidence, and node representation strategy.

Given the comparable accuracy of the methods examined in this survey, the selection of a specific method in a real-case scenario relies heavily on the application context of the AND. In the case of big data environments, like SKGs, where the user has access to extensive information derived from entity relationships (e.g., citations, research data links, project links, etc.), an approach that encompasses this input may be more advantageous than an approach that artificially constructs relationships through pairwise comparisons. Conversely, in environments where the entity collection is flat, hence links between entities are not readily available, constructing a graph based on pairwise comparisons is likely the best approach.

Generally speaking, methods based on unsupervised learning are the most popular because they mimic the real-case scenario of AND, where authors are often poorly described and persistent identifiers are not always available to create a ground truth to be used for the training of the learning architecture. A semi-supervised approach may be relevant to overcome such limitations by taking advantage of the little information provided by the collection together with the inference provided by the learning process.

To facilitate the evaluation of an AND method, numerous datasets and benchmarks have been developed. These resources are created by selecting an initial collection and then extracting a subset of “labeled” authors with verified identifiers (e.g., ORCIDs) whose metadata and context emulate the desired outcome of the method under testing. The labels assigned to this subset are subsequently utilized to determine the correctness of the disambiguation based on the equivalence of authors’ identifiers. Achieving good accuracy on this labeled subset ensures a comparable accuracy on the entire set of authors from which the subset has been extracted.

Many further developments are possible within this research scope, for example on explored evidence and node representations. SKGs are spreading in the domain of Open Science, addressing the demands for FAIR and reproducible science by including information relative to authors, publications, research data, research software, organizations, projects, grants, facilities, tools, instruments, and so on. Such SKGs offer new data evidence, hence new opportunities, to extend AND methods beyond the current publication-focused solutions. Another interesting direction is that of hybrid approaches, combining different node representation strategies of graph-based methods: pairwise similarity match can be used to define a similarity graph that can be consequently used as input to create node embedding derived from the neighborhood.

Additional Information and Declarations

Competing Interests

Author Contributions

Data Deposition

The authors declare that there are no competing interests.

Michele De Bonis conceived and designed the experiments, performed the experiments, analyzed the data, prepared figures and/or tables, and approved the final draft.

Fabrizio Falchi analyzed the data, prepared figures and/or tables, authored or reviewed drafts of the article, and approved the final draft.

Paolo Manghi analyzed the data, prepared figures and/or tables, authored or reviewed drafts of the article, and approved the final draft.

The following information was supplied regarding data availability:

Data is available at Zenodo:

Michele De Bonis. (2023). List of articles resulting from the Google Scholar search “graph based author name disambiguation” published after 1/1/2021 [Data set]. Zenodo. https://doi.org/10.5281/zenodo.8117573.

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
