# Peer review of "Graph-based methods for Author Name Disambiguation: a survey"

_PeerJ Computer Science, doi:10.7717/peerj-cs.1536_

## Round 0.1 · original submission · Minor Revisions

I am sure that you will be able to answer all comments/suggestions of the reviewers.

·

Basic reporting

This article sets out to review the important problem of disambiguating author names in scientific publications, that is mapping each author to a specific person, eg, disambiguating the different John Smith and J. Smith etc. More specifically the article focuses on techniques that are graph-based.

This reviewer certainly believes this is an important and open problem. Many subtly different techniques have been proposed and a coherent and up-to-date review is certainly very welcome.

Overall the introduction does a good job of describing the problem. The only addition we could recommend is that ORCID (https://orcid.org/) could be given at least some passing mention in the introduction?

Experimental design

One interrogation is the perimeter of the study itself. In section 3, line 153

"The articles selected for this survey have been found by searching for the keywords “graph-based author name disambiguation” on Google Scholar and filtering those proposing deep learning approaches"

Should this specific choice of deep learning be motivated? Are there graph-based but non deep learning studies that have been ignored? Or is the entire field only focused on deep learning in the past few years anyway? This could be clarified and maybe even added all the way to the title of the article?

The authors could be more precise and somewhat expand that section 3 "survey methodology". When trying to reproduce as exactly as possible what is described one ends up with that search:

https://scholar.google.com/scholar?q=graph-based+author+name+disambiguation&as_ylo=2020

As of 2023-05-23, that returns 3 680 articles. It seems difficult to believe that all of them were inspected one by one (or even their titles scanned) for mentions of "deep learning". Therefore some other heuristic must have been used? What is it exactly? Some brief clarification would be welcome.

Validity of the findings

Overall the different techniques are nicely summarized in coherent paragraphs. Table 2 and 3 offer a good simplified overview and the article does a good job of providing the reader with a good recent review of the field as well as a useful taxonomy.

The conclusion does mentions interesting possible future directions. The sentence "Nevertheless, existing surveys on the topic do not... line 421" reads a bit too much like self-justification and could probably be re-written in a more positive tone to simply re-iterate what the paper brings.

A useful paper that would certainly be well received by the community!

Additional comments

some really minor syntactic or stylistic pointed noted by this reviewer (who is not a native english speaker)

line 28
for discovery, assessment, and statistics and trends.
-> remove and

line 45
Several methods have been proposed ed engineered
-> and

line 274
check capitalization and maybe provide link to dataset / competiion?
(appears to be: https://www.biendata.xyz/competition/whoiswho1/)
is that the same as footnote 8?

footnote 11
fix: https://citeseerx.ist.psu.edu/index -> https://citeseerx.ist.psu.edu/
(please check all the links in the article)

Style and readability could overall be improved slightly. Maybe some good editor could provide some help? Or, if that is allowed, some careful use of a language model for inspiration on possible rewrites of some punctual sentences?

Cite this review as

·

Basic reporting

The article surveys the current state-of-the-art in Author Name Disambiguation based on scholarly knowledge graphs. The topic of the survey is of general interest and within the scope of the journal. The authors report on several related surveys on the more general topic of Author Name Disambiguation, but clearly explain the added value of their work with respect to scholarly knowledge graphs. Moreover, as the field is actively researched,, the survey fill an open gap.

The article is written in a nice and thorough way and is easy to follow. However it contains some typos and similar. The authors should carefully revise the document.
* l45: "ed"
* When citing multiple reference in a row, they should be separated by commas, for instance l357, l389
* l304: "i.g."
* l322: "a SKG" -> an SKG
* l329 an l330: semantic features and semantic features (I think one should be node the other relation semantic features)
* l332: "the SKGs is"
* l398: "a pair of author names are" -> is
In addition, the manuscript contains some issues with spaces, as for instance in l326 "SKG : ", l329 "i.e. " (it looks like a space at the and of the sentence issued by latex due to the ".")

The tables could be improved.
* Table 1, column 1 contains the author name twice and the year in column 1 and 2. To remove redundant information, the citation (incl. author name and year) is sufficient
* Table 2 and 3 could be extended by the citation helping the reader to understand the presented information. Also the width of the columns could be adjusted to the contained information.

Figure 1 provides a generalized overview all AND methods surveyed. Some text in the figures is to small to read.

When it comes to datasets, I would recommend the authors to include formal citations to the datasets instead of just links in footnotes. This provides credit for the authors of those dataset.

In addition to the above described issues, I have problems understanding these text passages:
l337: What is meant by "intent-driven views"? please clarify
* l362: What is meant by "co-authored one application" Is it application in the sense of software? Nothing in this direction was mentioned before. Please clarify*

Experimental design

The survey selected articles based on their citation count from google scholar search for the presented keywords. While this reduces the number articles to be reviewed, I have the feeling that this selected introduces a bias toward highly cited (and potentially already included in other surveys) articles. Furthermore, this might exclude recently published articles. The authors should ensure to prevent this bias. Furthermore, I would like to see the exact date of the query and the number of results provided by the databases. With respect to reproducibility, It would be great if the authors attach a list of all results as supplementary material to the article.

Beside approaches to AND, they authors claim to review existing benchmark datasets, but provide only little information about the datasets. A table could be included that compares several properties, such as number of publications, authors, venues, range of years covered, best current result for AND, and others. This would help readers to actually understand the differences between the datasets.

I see two general reasons for surveys: (1) help researchers to find open research topics and (2) help researchers to actually select an approach particularly suited for a problem at hand. While the survey does a good job for (1), I'm missing some information with respect to (2). This includes benchmarks results of the different approaches as well as a concluding summary that provides some help with respect to deciding upon the approach to select.

Validity of the findings

The findings seem to be valid, but the selection of the literature might have introduced a biased as described above.

Additional comments

I would like to see the survey published, but I think the authors should improve the selection as well as the reporting w.r.t. the datasets.

Cite this review as

---

## Round 0.2 · accepted · Accept

I confirm that the authors have addressed all of the reviewers' comments. I hope that the authors will have an opportunity to correct two references as reviewer 1 has proposed recently.

·

Basic reporting

The authors have addressed the initial reviewer suggestions and the article is definitely stronger and clearer for it.

Experimental design

The re-write of section 3 was particularly important. One minor hiccup is that the actual list of papers considered in the review is cited as "De Bonis, M. (2023). List of articles resulting from the Google Scholar search ”graph based author name disambiguation” published after 1/1/2021". This does not appear to be a valid citation that anyone could just access. Would it not be possible to find a way to add this CSV file as supplementary online material somehow? Or host it somewhere as a dataset with a proper URL?

Validity of the findings

(no further comments in this 2nd reviewing round)

Additional comments

Line 528: the acronym FAIR should be defined.

Cite this review as

·

Basic reporting

The authors addressed all of my comments.

Experimental design

The authors addressed all of my comments.

Validity of the findings

The authors addressed all of my comments.

Cite this review as